# Hypomethylation of the *RUNX2* Gene Is a New Potential Biomarker of Primary Osteoporosis in Men and Women

**DOI:** 10.3390/ijms25137312

**Published:** 2024-07-03

**Authors:** Bulat Yalaev, Anton Tyurin, Karina Akhiiarova, Rita Khusainova

**Affiliations:** 1Endocrinology Research Centre, Dmitriya Ulianova Street, 11, 117036 Moscow, Russia; yalaev.bulat@yandex.ru (B.Y.); ritakh@mail.ru (R.K.); 2Internal Medicine & Clinical Psychology Department, Bashkir State Medical University, 450008 Ufa, Russia; liciadesu@gmail.com; 3Medical Genetics Department, Bashkir State Medical University, 450008 Ufa, Russia

**Keywords:** osteoporosis, methylation, pyrosequencing, CpG, *RUNX2*

## Abstract

The search for the molecular markers of osteoporosis (OP), based on the analysis of differential deoxyribonucleic acid (DNA) methylation in bone cells and peripheral blood cells, is promising for developments in the field of the early diagnosis and targeted therapy of the disease. The Runt-related transcription factor 2 (*RUNX2*) gene is one of the key genes of bone metabolism, which is of interest in the search for epigenetic signatures and aberrations associated with the risk of developing OP. Based on pyrosequencing, the analysis of the *RUNX2* methylation profile from a pool of peripheral blood cells in men and women over 50 years of age of Russian ethnicity from the Volga-Ural region of Russia was carried out. The level of DNA methylation in three CpG sites of the *RUNX2* gene was assessed and statistically significant hypomethylation was revealed in all three studied CpG sites in men (U = 746.5, *p* = 0.004; U = 784, *p* = 0.01; U = 788.5, *p* = 0.01, respectively) and in one CpG site in women (U = 537, *p* = 0.03) with primary OP compared with control. In the general sample, associations were preserved for the first CpG site (U = 2561, *p* = 0.0001766). The results were obtained for the first time and indicate the existence of potentially new epigenetic signatures of *RUNX2* in individuals with OP.

## 1. Introduction

Primary osteoporosis is one of the most common age-related metabolic diseases, leading to bone fragility and a high frequency of fractures. The disease is multifactorial and clinically heterogeneous, with complex disruptions in the molecular control of bone cell differentiation, as well as the endocrine regulation of catabolic and anabolic processes in the musculoskeletal system underlying its pathogenesis. Genetic and epigenetic factors significantly contribute to the risk of disease development [1,2]. According to recent estimates, more than 18% of the world’s population is affected by osteoporosis [3]. In Russia, osteoporosis is diagnosed in approximately 34% of the women and 24% of the men [4]. Despite all the healthcare measures aimed at reducing morbidity and lowering diagnostic costs, the global number of fractures and subsequent post-traumatic complications is increasing. It is expected that by 2050, the frequency of hip fractures in men worldwide will increase by 310%, and in women by 240% [5]. This is due to the increasing proportion of the elderly population and the lengthy asymptomatic course of the disease, which is usually diagnosed only after the first fractures. Currently, osteoporosis diagnosis is based on bone mass assessment using dual-energy X-ray absorptiometry (DEXA) and fracture risk calculation using online calculators such as FRAX [6]. However, the effectiveness of bone densitometry and risk assessment is insufficient for early disease diagnosis. This is a current issue in practical healthcare, the solution of which requires a comprehensive approach using genetic technologies [7].

The solution to this problem lies in the search for molecular/genetic and epigenetic markers that can serve as the early predictors of osteoporosis at a fundamental level, serving as the precursors and markers of the onset of pathogenetic processes leading to an imbalance in bone remodeling, and ultimately leading to the development of this disease [8]. From this point of view, it is advisable to search for the early markers of osteoporosis among the primary regulatory links, factors of bone metabolism, and subtle changes in anabolic processes in bone tissue.

Transcription factor 2 (*RUNX2*) is a transcription factor 2 associated with RUNT, a key regulator of osteoblast differentiation and the cell cycle in general [9]. It acts as the primary control point in the chain of mesenchymal stem cells (MSCs) entering the osteoblastogenesis pathway, being equally important in functionality to the transcription factor SP7 and the WNT signaling. The *RUNX2* gene consists of 10 exons and has 12 transcripts (splice variants) [10,11], and is highly expressed in the early stages of bone cell development when mesenchymal stem cells differentiate into osteoblast precursors [11]. However, its production significantly decreases in mature osteoblast stages. In the G1 phase of the cell cycle in the MC3T3-E1 osteoblast cell line, the level of *RUNX2* is maximal and minimal during G2, S, and mitosis [12]. The transcription factor acts in multiple directions, inducing the activity of multiple genes specifically expressed in osteoblast lines, including genes *OSX* (osterix), *OCN* (osteocalcin), and *BSP* (bone sialoprotein), and suppresses the expression of the genes that decrease osteoblastogenesis, such as PPAR-γ (peroxisome proliferator-activated receptor gamma), and *MyoD* (myogenic differentiation), which functionally aim at enhancing the adipogenic and myogenic pathways of the mesenchymal stem cell differentiation [13,14,15].

Several epigenetic mechanisms are known to regulate the expression of genes controlled by *RUNX2*. In particular, *RUNX2* exhibits activity in mitotic chromosomes as a factor in epigenetic induction to maintain and preserve cellular identity after mitosis [16,17]. The contribution of *RUNX2* to the epigenetic regulation of gene expression during osteoblast differentiation has been demonstrated through interactions with histone deacetylases [18], histone acetyltransferases [19], and components of the SWI/SNF complex [20]. It is known that the inhibition of ovulation may lead to an increase in the methylation status of the *RUNX2* promoter in bone, suppress its transcription, reduce translation, and consequently increase the risk of developing osteoporosis [2]. Mice with a *RUNX2* knockout develop cleidocranial dysplasia syndrome, while mice with overexpression experience disruption in bone mineralization, indicating the complex nature and critical role of this transcription factor in bone tissue metabolism. Research shows that the expression of the *RUNX2* gene is lower in the circulating mesenchymal stem cells of osteoporosis patients compared to a control group [21]. *RUNX2* is regulated by various factors. From the perspective of epigenetic regulators, microRNAs regulating *RUNX2* are well studied. For example, *miRNA-194* modulates mesenchymal stem cell differentiation [22] and accelerates osteoblast differentiation through the nuclear translocation of *RUNX2* by the STAT1 [23] signaling transducer. *miR-133a-5p* inhibits the expression of the *RUNX2* gene at the transcriptional and translational levels by binding to the 3′-untranslated region of *RUNX2* mRNA [24]. The phosphorylation of *RUNX2* mobilizes chromatin regulatory factors and accelerates mesenchymal stem cell maturation. For instance, by phosphorylating specific serine residues at positions 301 and 319, *RUNX2* induces osteocyte maturation through MAPK-dependent signaling [23,25] and BMP2-sensitive transcription [26]. The transcriptional activity of the *RUNX2* gene is enhanced by the acetyltransferase p300 and nicotinamide phosphoribosyltransferase (NAMPT), which in turn promotes the osteogenic differentiation of MSCs [27]. Literature analysis provides a fairly good understanding of how *RUNX2* is regulated through epigenetic mechanisms; however, it only provides a superficial understanding of whether aberrations in these mechanisms are directly linked to the risk of developing primary osteoporosis, and whether these changes can be identified, particularly in the methylation profile overall and at the individual CpG sites of the *RUNX2* gene in individuals with osteoporosis.

The differential DNA methylation of the promoters and regulatory regions of the *RUNX2* gene is poorly understood in the context of associative search with osteoporosis itself, as well as with individual endophenotypes: fractures and low BMD levels. The whole-epigenome studies (EWAS) of osteoporosis have not identified the significant signatures of the differential CpG site methylation of the *RUNX2* gene [28], while according to various data, the methylation status of the gene may influence the risk of ankylosing spondylitis [29], and the methylation status of the adjacent region to the *RUNX2* gene may be associated with the risk of developing osteoarthritis [30]. According to Haga et al. (2015), the methylation status of the promoter region of *RUNX2* did not change during osteoblastic differentiation; however, the analysis was only conducted on the MSC cultured line without reference to the osteoporotic phenotype [31]. Thus, the role of the differential and aberrant methylation of the *RUNX2* gene in individuals with osteoporosis remains an unresolved scientific problem that requires further research and the search for DNA methylation status signatures in this gene that may be directly associated with the osteoporotic phenotype, i.e., the risk of fractures and low BMD levels.

The aim of the study: The assessment of methylation status at three CpG sites in the promoter region of the *RUNX2* gene in women and men over 50 years with fractures and low bone mineral density from the Volga-Ural region of Russia.

## 2. Results

### 2.1. Associative Analysis

An analysis of the DNA methylation of the *RUNX2* gene was conducted considering the mean methylation level across three CpG sites. Table 1 presents the data of the mean values and medians of the DNA methylation level in the studied sequence of the gene.

All the launches and readings of the sequences during pyrosequencing passed automated quality control. Figure 1 shows the pyrogram of one of the samples. Using the RStudio package libraries, a statistical evaluation of significant differences in methylation status between the control group and individuals with osteoporosis was first conducted based on the Mann/Whitney criterion, both for the overall sample and separately for the samples of women and men. Statistically significant hypomethylation was identified in the *RUNX2* gene at all three CpG sites in men (U = 746.5, *p* = 0.004; U = 784, *p* = 0.01; U = 788.5, *p* = 0.01, respectively) and at one CpG site in women (U = 537, *p* = 0.03) with primary osteoporosis (men and women combined) compared to the control sample (Figure 2). When combining the samples of men and women, there remains a statistically significant association of hypomethylation at the CpG1 site in individuals with OP compared to the control, showing a higher level of significance (U = 2561, *p* = 0.0001766), and a significant association is also found for the average level of the methylation of all three CpG sites (U = 3117, *p* = 0.04496), Figure 3.

Table 2 shows the data of logistic regression analysis, which confirm the data of non-parametric analysis: in prediction models for men, the hypomethylation of 1, 2, and 3 CpG sites, as well as their average value, is a statistically significant predictor of osteoporosis compared to the control sample; in prediction models for women, the hypomethylation of only 1 CpG site is statistically significant. In the analysis of the logistic model in the total sample, the hypomethylation of CpG site 1 shows the highest level of statistical significance, while the average value does not demonstrate significant differences.

### 2.2. Functional Significance of Methylation in the Studied CpG Sites of RUNX2

We analyzed the structure of the studied sequence in the publicly available databases UCSC Genome Browser on Human (GRCh38/hg38), NCBI, Ensembl, and ClinVar to assess the functional significance and likely consequences of the changes in the methylation status of the studied CpG sites associated with risk phenotype.

The studied DNA fragment at genomic position 45420262-45420295 is located on the reverse strand between exon 2 and 3, 2297 nucleotides from exon 3 (Figure 4).

According to the Ensembl database, referencing Havana annotation data, the studied region is located in the promoter region of the *RUNX2* gene. Downstream in 262 nucleotides is the transcription repressor CTCF or CCCTC-binding factor. Therefore, the studied DNA fragment is a regulatory region involved in transcription processes and gene expression regulation.

The entire fragment under investigation, according to the UCSC browser database, is localized in an enhancer-like region (proximal enhancer-like signature). Such enhancers typically exhibit a high level of the histone mark H3K27ac, which in turn indicates the acetylation of the lysine residue at position 27 in the N-terminal of the histone H3 protein. H3K27ac is associated with higher transcriptional activation and is defined as a mark of active enhancers. Most CpG islands in the genome are known to be located outside regulatory regions. Active promoters are predominantly unmethylated regions (UMRs) with methylation levels below 20%, while cis-regulatory sequences such as enhancers demonstrate methylation levels in the range of 20% to 80% [32]. Therefore, it can be assumed that hypomethylation in the investigated DNA segment likely enhances the function of this enhancer-like region, leading to enhanced gene transcription; however, further functional studies are needed.

The first CpG site in the examined sequence is located in the intronic region at position 45420291-45420292. This region is not described in ClinVar, according to the NCBI and Ensembl databases, where the single nucleotide polymorphic variants rs993615980 and rs1380047352 are located, which are not described as pathogenetically significant. In the region where 2 and 3 CpG sites are located, the polymorphic variants rs1798151552, rs907471189, and rs1798151457 are found, for which there are also no data from functional studies.

## 3. Discussion

It has been established for the first time that the gene *RUNX2* in the promoter region is hypomethylated in postmenopausal women with osteoporosis at 1 CpG site, while in men over 50 years old with osteoporosis, all three studied CpG sites are hypomethylated, indicating the significant role of the aberrant DNA methylation of the *RUNX2* gene in the development of this disease. Currently, there are no studies describing a decrease in the methylation status of this promoter region of the *RUNX2* gene in patients with osteoporosis, so this result is obtained for the first time. The results are scientifically novel; however, validation is required in independent samples.

It is known that alterations in the methylation profile of the SOST gene disrupt the transactivation of *RUNX2* in postmenopausal osteoporosis patients and affect bone tissue metabolism [33]. It has been shown that it is regulated by histone deacetylases; in particular, HDAC4 directly deacetylates *RUNX2*, suppressing its transcriptional activity and promoting its degradation in mature osteoblasts [34]. Thus, it can be assumed that the hypomethylation of the promoter region of the *RUNX2* gene and changes in its expression level may be associated with chromosomal remodeling through the epigenetic mechanisms of histone acetylation.

Rice et al. (2018) identified the regulatory region of the *RUNX2* gene, whose methylation status was found to be associated with osteoarthritis (OA). By conducting a series of experiments with cells from the joints of 260 OA patients in vitro, including using CRISPR-Cas9, the authors of the study investigated single nucleotide polymorphisms located within the differentially methylated region of *RUNX2* and localized in this region SNP associated with OA, and found that rs10948172 showed a strong correlation with the level of methylation of *RUNX2*, and two intergenic SNPs, falling in the methylation region (rs62435998 and rs62435999), based on functional studies, showed genetic and epigenetic effects on the regulatory activity of this region [30].

Recently, Wong et al. (2023) identified differences in the promoter methylation of the *RUNX2* gene and its transcriptional level in ankylosing spondylitis (AS). Four CpG regions and 74 CpG sites of *RUNX2* were studied, among which CpG-2, CpG-4, and 18 other CpG sites were differentially methylated. The methylation of CpG-4 sites negatively correlated with C-reactive protein (*p* < 0.05) in AS patients. In the qRT-PCR validation stage, the mRNA level of *RUNX2* in AS patients was significantly higher compared to the control group (*p* < 0.05), and in AS patients receiving biological agents, the methylation level of CpG-2 sites showed a negative correlation with mRNA levels (*p* < 0.05). The ROC results demonstrated that the methylation of the *RUNX2* gene and its transcriptional level have a good potential to distinguish AS patients from HCS [29].

Krstic et al. (2022) have demonstrated in the study on a mouse model that vitamin D deficiency in early age modulates changes in the methylation of the gene *RUNX2* promoters in the tibia bone subjected to mechanical loading. In the loaded limbs of mice receiving a prenatal diet deficient in vitamin D, the methylation of *RUNX2* at CpG site 24 was lower compared to control mice. The CpG sites of *RUNX2* are located 2 kb upstream from the transcription start site of *RUNX2*, and Ensembl displays this region as part of the *RUNX2* promoter. Wakitani et al. found there is a reverse correlation between the DNA methylation of *RUNX2* CpG-2101 and the expression of the *RUNX2* gene. This CpG site is located 40 bp away from the CpG sites of interest authors, therefore, similar associations are likely to be shown. This suggests that lower methylation is likely associated with the increased expression of the *RUNX2* gene [35].

In the cluster of scientific works aimed at finding biomarkers for osteoporosis, the focus is gradually shifting towards epigenetic research. DNA methylation, RNA interference, and the post-translational modifications of histones are not only candidates for the markers of fractures or low BMD levels in osteoporosis [36]. They are also attractive molecular targets for the development of gene therapy and target drugs [37]. Significant progress has been made in this field. Epigenetic therapeutic agents have already been developed, which can be classified into several groups based on the mechanism of action: drugs primarily targeting epigenetic enzymes, including the inhibitors of DNA methyltransferases (DNMTs), histone acetylation (HAT), histone deacetylation (HDAC), histone demethylation (KDM), and epigenetic reader blockers containing bromodomains (BRDs). Compounds known to act on the epigenetic regulation of bone tissue remodeling balance include immunomodulators targeting the NF-κappaB ligand receptor involved in RANKL and SOST acetylation, which are involved in bone tissue metabolism [38]. Specifically, butyrate stimulates histone H3 acetylation, the production of 8-isoprostane, and RANKL expression, and regulates osteoprotegerin expression/secretion in MG-63 osteoblastic cells [39]. As for NFkappaB, it is associated with chromatin decondensation through the disruption of nucleosome histone-DNA interactions, providing the activation of hidden enhancers that modulate immune response gene expression. Thus, the temporal dynamics may determine the ability of the transcription factor to reprogram the epigenome in a stimulus-specific manner [40]. Compounds that act on the epigenetic landscape of bone tissue remodeling balance are known, such as immunomodulators targeting the NF-κB ligand receptor activator, which affects acetylation (RANKL) and (SOST), as well as agents that affect the substrate of epigenetic enzymes, such as bisphosphonates (BPSs), which target the metabolic pathway of bone genesis [41]. 

More than 150 types of RNA modifications have been identified, among which N6-methyladenosine modification is the most common modification in mammalian cells, occurring in the adenosine base at the nitrogen-6 position of mRNA. Unlike other gene modifications, m6A modification is dynamically reversible. Recent studies have shown that m6A methylation is involved in the development of bone diseases such as osteoporosis and osteoarthritis. Yan et al. (2019) found that the knockout of the gene encoding methyltransferase-like protein 3 (METTL3) in humans and mice blocked the m6A methylation of mRNA and precursor miR-320. According to the authors’ preliminary conclusions, METTL3 acts as an anti-osteoporotic factor or pro-osteogenic factor, at least in part, by maintaining the expression of the *RUNX2* gene at a higher level through the dual mechanisms of the direct m6A methylation of *RUNX2* and indirect upregulation of the *RUNX2* level by the methylation of pre-miR-320 [42].

However, the application of epigenetic therapeutic agents is associated with a number of difficulties, including insufficient data on the specificity of action, toxicity, and individual tolerability. On the other hand, in terms of searching for the early epigenetic markers of osteoporosis, there is a problem of the lack of scientific data on interindividual, interpopulation, and age-related differences in DNA methylation patterns in a number of candidate genes involved in the pathogenesis of the disease. The reversible nature of DNA methylation, age-related characteristics, as well as the complex mechanisms of the interaction of methylated genes with the molecular microenvironment and the specificity of interaction with other genes make candidate epigenetic studies of osteoporosis no less important than whole epigenome studies, as they allow both the replication of the existing data and a more focused approach to studying the DNA methylation of genes involved in bone metabolism.

Cultivating osteoblasts and osteoclasts in vitro can alter the original patterns of epigenetic marks, distancing experimenters from an accurate representation of the initial intracellular processes of the epigenetic regulation of osteogenic cells in vivo [43]. Therefore, the search for reliable biomarkers among epigenetic regulators should be focused either on cells in tissues where epigenetic patterns are sufficiently stable and these cells are readily accessible for study, or the search for such markers should be focused on cells from peripheral blood, which are more accessible for laboratory screening. The comprehensive study by Ebrahimi et al. (2021) convincingly demonstrates that the DNA methylation status in peripheral blood cells may reflect bone-specific methylome, allowing the identification of CpG sites associated with the regulation and dysregulation of bone tissue metabolism [43]. Thus, peripheral blood is an obvious choice as a non-invasive substitute for developing accessible methods of epigenetic DNA diagnostics.

## 4. Materials and Methods

### 4.1. Study Sample

The “case/control” study involved 96 postmenopausal women (mean age 61.95 ± 7.94, mean weight 60.15 ± 7.43, BMI 27.0 ± 2.40) and 96 men (mean age 62.00 ± 10.8, mean weight 63.24 ± 9.51, body mass index 27.48 ± 3.60) of Russian ethnicity, examined at the Bashkir State Medical University. The sample included women (N = 48) and men (N = 48) with primary osteoporosis, and the control comparison group consisted of individuals without fractures and with normal BMD levels (women: N = 48; men: N = 48). Relatives were excluded from the sample. Additionally, exclusion criteria included a history of alcohol and drug abuse, smoking, long-term use of glucocorticoids, and hormone replacement therapy. BMD levels were measured using dual-energy X-ray absorptiometry (DEXA) with the Hologic QDR 4500/A DXA system (Marlborough, MA, USA) at standard locations (hip and lumbar spine). The overall sample was divided according to the T-score criterion—from +2.5 to −0.9 standard deviations (SDs) indicated normal BMD, values from −1.0 to −2.5 SD indicated osteopenia, and values below −2.5 indicated osteoporosis (according to the World Health Organization recommendations). The presence of osteoporotic fractures in standard locations (hip axis and lumbar spine) in general and individually, as well as in combination with any other skeletal fractures, was also considered in the patients. Each participant signed an informed consent form to participate in the study in accordance with the standards of the Helsinki Declaration of the World Medical Association “Ethical Principles for Medical Research Involving Human Subjects”.

### 4.2. DNA Methylation Analysis

The selection of 3 CpG sites in the *RUNX2* gene was carried out based on their absence in the early epigenetic studies of osteoporosis as research targets, as well as based on the automatic primer selection mode using the patented technology for CpG island analysis from QIAGEN^®^ (Hilden, Germany), aimed at generating the most optimal primer design, via the GeneGlobe web interface (URL: https://geneglobe.qiagen.com/ru/customize/pyrosequencing/pyromarkcpgandarrayvalidationassays, access on 18 October 2020). Table 1 provides information on the studied sequence.

DNA extraction was performed using the phenol/chloroform extraction method from peripheral blood leukocytes according to the Mathew et al. protocol (1984) [44]. The quality of the extracted DNA was checked using a NanoDrop 1000 spectrophotometer (Thermo Scientific, Waltham, MA, USA). The DNA concentration was measured using a Qubit 4 fluorometer (Thermo Scientific, USA). For the analysis of the methylation profile of the *RUNX2* gene, the bisulfite conversion method was applied to the original DNA, followed by pyrosequencing on the Pyromark Q24 platform (QIAGEN^®^, Germany).

The research consisted of several consecutive stages:The bisulfite conversion of the DNA samples was performed using the EpiTect Fast DNA Bisulfite Kit (QIAGEN^®^, Germany).The purification of the bisulfite-converted DNA was carried out on MinElute spin columns using the EpiTect Fast DNA Bisulfite Kit (QIAGEN^®^, Germany).The polymerase chain reaction (PCR) of the converted DNA was performed using 2 primers, one of which was biotinylated (PyroMark PCR Kit (QIAGEN^®^, Germany)).Pyrosequencing with the sequencing primer was performed using the PyroMark Gold Q24 Reagents Kit (QIAGEN^®^, Germany).

The design of the primers flanking the gene regions with methylated CpG sites was constructed on the GeneGlobe web platform (QIAGEN^®^, Germany). The inclusion of a single biotinylated PCR primer allows for the separation of the two amplicon strands to create a matrix of ssDNA for annealing the pyrosequencing primer and extending the complementary strand by the discrete distribution of nucleotides.

Table 3 presents the analyzed region of the gene under study. An intronic region of the *RUNX2* gene, located between positions 45420262 and 45420262 in the genome, containing 3 CpG sites, was selected as the region of interest. The length of the amplicon that was subsequently sequenced was 99 nucleotides. For the quality control of the sequencing, demethylated (0%) and methylated (100%) DNA control samples from the manufacturer (QIAGEN^®^, Germany) were included in the number of samples analyzed.

### 4.3. Statistical Analysis

During the pyrosequencing work, the light trace for each well detected by the camera was displayed in real-time mode, generating peak pyrograms, the height of which indicated the stoichiometric inclusion of nucleotides. Each peak without CpG becomes a reference peak, which the software uses to calculate the percentage of sample methylation. All the starts and reads of the sequences during pyrosequencing have passed automatic quality control. First, the statistically significant differences in the methylation status between the control group and individuals with osteoporosis were evaluated using the R Studio library package based on the nonparametric Mann/Whitney criterion at a significance level of *p* < 0.05 adjusted for continuity. A non-parametric Mann/Whitney criterion was used to detect the statistically significant differences between the control group and individuals with OP at a significance level of *p* < 0.05 with a continuity correction. Logistic regression analysis was conducted to identify the predictors of OP among the analyzed CpG sites. In this case, the disease phenotype served as the dependent variable, and the measured percentage of the methylation of the analyzed regions (in CpG islands) served as the independent variables (predictors) at a significance level of *p* < 0.05. The analyses were performed using the Rstudio software with packages for statistical analysis based on the non-parametric criteria and logistic regression analysis.

## 5. Conclusions

Thus, the statistically significant hypomethylation of the three studied CpG sites in the promoter region of the *RUNX2* gene was first detected in men and in one CpG site in the promoter region in women with primary osteoporosis compared to the control group. Moreover, in the combined sample of men and women, significant differences with a high level of significance are maintained for the first CpG site of *RUNX2*.

## Figures and Tables

**Figure 1 ijms-25-07312-f001:**
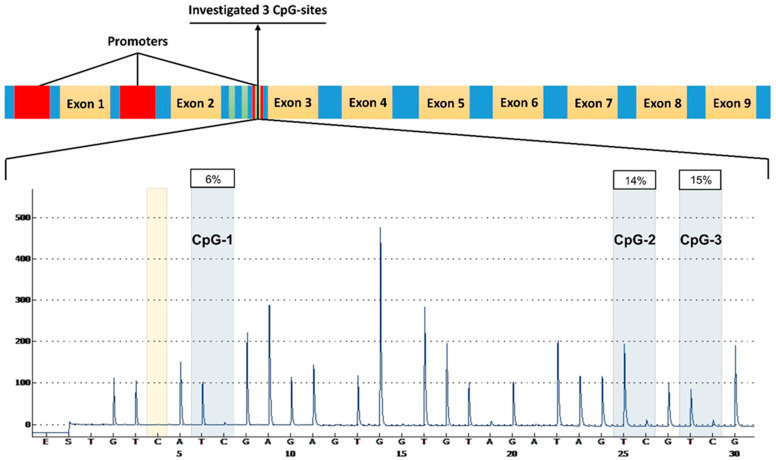
The top part illustrates the structure of the gene *RUNX2* schematically, indicating the localization of the investigated CpG sites. At the bottom, a pyrogram is shown, visualizing the peaks of nucleotide sequence readings with integrated information about the percentage of the methylation of the investigated CpG sites. The human genome assembly GRCh38. The CpG sites that pass quality control are marked in blue, while those that do not pass are marked in red. The yellow band represents internal control for nucleotide sequence conformity.

**Figure 2 ijms-25-07312-f002:**
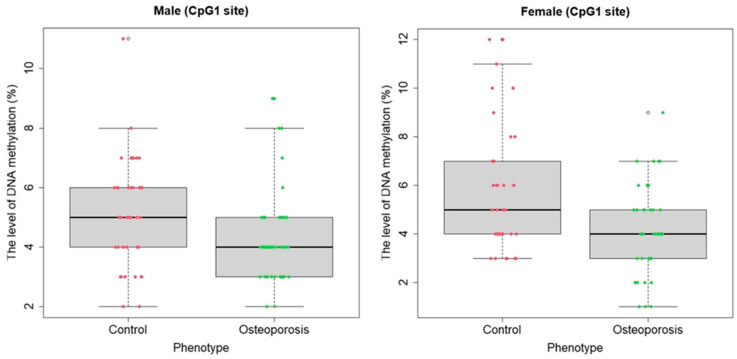
Distribution graph of the methylation levels (status) of the CpG-site of the *RUNX2* gene in the comparison groups of men and women (mean values and percentage confidence intervals are shown). On the abscissa axis, there is a separation by the presence/absence of the disease; on the ordinate axis, there are the indicators of the average methylation level and confidence intervals (in percentages).

**Figure 3 ijms-25-07312-f003:**
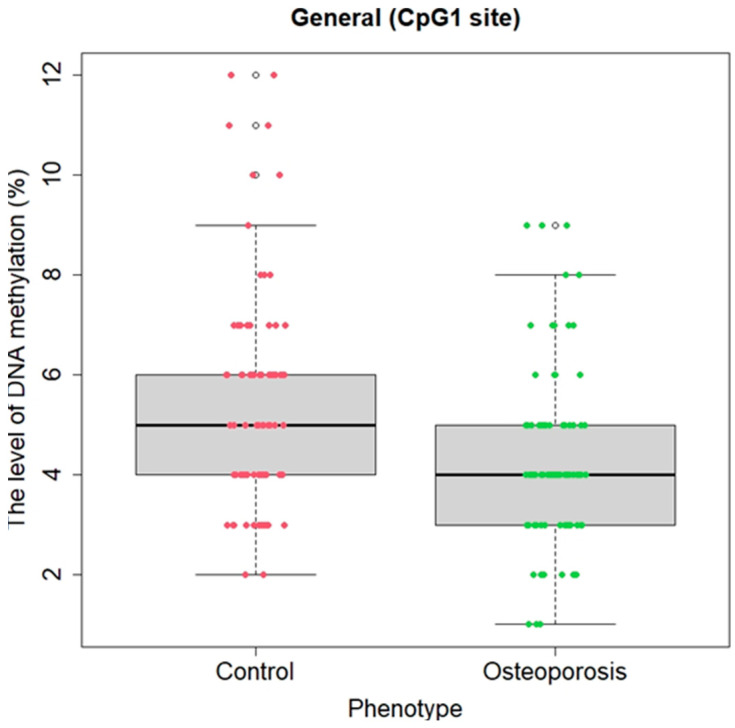
Distribution graph of the methylation levels (status) of the CpG site of the *RUNX2* gene in the comparison groups in the general sample (mean values and percentage confidence intervals are indicated). On the abscissa axis, there is a separation by the presence/absence of the disease; on the ordinate axis, there are the indicators of the mean methylation level and confidence intervals (in percentages).

**Figure 4 ijms-25-07312-f004:**
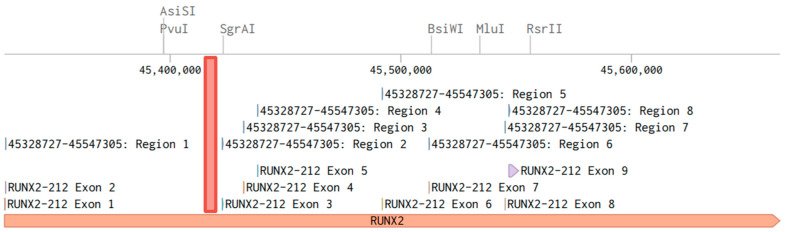
Structure of the *RUNX2* gene; the investigated DNA section is highlighted in red. The screenshot taken from the benchling.com web platform. GRCh38 human genome assembly.

**Table 1 ijms-25-07312-t001:** Average arithmetic values and medians of methylation level of *RUNX2*.

Females							
Median (case) (%)				Median (control) (%)			
CpG1	CpG2	CpG3	Mean	CpG1	CpG2	CpG3	Mean
4	8	9	7	5	8	9	7.67
Mean (case) (%)				Mean (control) (%)			
CpG1	CpG2	CpG3	Mean	CpG1	CpG2	CpG3	Mean
4.21	9.51	10.08	7.93	5.7	9.35	9.02	8.03
Males							
Median (case) (%)				Median (control) (%)			
CpG1	CpG2	CpG3	Mean	CpG1	CpG2	CpG3	Mean
4	8	9	7	5	10	10	8.33
Mean (case) (%)				Mean (control) (%)			
CpG1	CpG2	CpG3	Mean	CpG1	CpG2	CpG3	Mean
4.34	8.48	9.18	7.35	5.20	9.82	10.51	8.51
General sample							
Median (case) (%)				Median (control) (%)			
CpG1	CpG2	CpG3	Mean	CpG1	CpG2	CpG3	Mean
4	8	9	7.33	5	9	9	7.67
Mean (case) (%)				Mean (control) (%)			
CpG1	CpG2	CpG3	Mean	CpG1	CpG2	CpG3	Mean
4.59	9.23	9.89	7.90	5.05	9.57	9.90	8.17

**Table 2 ijms-25-07312-t002:** Results of logistic regression analysis, which included methylation levels of CpG sites 1, 2, and 3 and their mean values in males.

Parameters	Estimate	Std. Error	z-Value	*p*-Value
Males				
Sample	1.632	0.749	2.178	0.029
CpG1	−0.325	0.152	−2.130	0.033
CpG2	−0.174	0.089	−1.941	0.042
CpG3	−0.204	0.093	−2.188	0.028
Mean	−0.234	0.110	−2.127	0.033
Females				
Sample	1.653	0.714	2.315	0.020
CpG1	−0.337	0.136	−2.466	0.013
CpG2	−0.156	0.132	−1.179	0.238
CpG3	0.300	0.166	1.807	0.070
Mean	−9.017	2.135	−1.990	0.067
General sample				
Sample	1.798	0.535	3.356	0.0008
CpG1	−0.370	0.108	−3.421	0.0006
CpG2	1.741	3.879	0.910	0.363
CpG3	1.859	3.865	0.916	0.360
Mean	−0.079	0.067	−1.184	0.236

Note: Estimate is the regression coefficient, Std. Error is the standard error, z-value is the standard deviation index, and *p*-value is the significance level.

**Table 3 ijms-25-07312-t003:** Characteristics of sequences for DNA methylation analysis used in this work.

Gene	Genomic Coordinates	Analyzed DNA Sequence	Analyzed DNA Sequence Sequence after Bisulfite Conversion	CpG- Sites
*RUNX2*	45420262-45420295	GCACGGAAGATGGGGGCCTGGTGCCAGTCGCGGA	GTAUGGAAGATGGGGGTTTGGTGTTAGTUGUGGA	**3**

## Data Availability

Data is contained within the article.

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
