# Peer review of "Hypomethylation of the RUNX2 Gene Is a New Potential Biomarker of Primary Osteoporosis in Men and Women"

_ijms, 2024, doi:10.3390/ijms25137312_

Round 1

Reviewer 1 Report

Comments and Suggestions for Authors

The subject of epigenetic regulation of bone metabolism is one of great interest and the investigation of differential methylation of RUNX2 and its potential role in the development of osteoporosis is worthy of study.

In the introduction the authors give a very good overview of the literature, although the structure could be improved, and missing is a clear rational for performing the study and for the selection of the three CpG sites studied. Improving the written descriptions and visual representations of the RUNX promoter region and location of the CpG sites in relation to other CPG sites, regulatory elements or SNPs would greatly improve the clarity. The discussion also provides a good overview of the literature but could be improved by contextualising and interpreting the results.

Specific comments include:

·     The title is, I think, overstated and revision is suggested.

·     Results:

o    Some methods are included here and would be more appropriate, moved to the relevant section.

o    Legends to figures and tables could be more detiled to aid understanding of the contents e.g. In table 2 what is the sample estimate?

o    Figure 1 resolution is poor. Figure 4 is not very informative. Inclusion of a diagram illustrating the CpG sites in relation to key features would aid interpretation.

·     Methods:

o    Description of the RUNX CpG sites, location etc should be included here to improve overall understanding. For example, it is unclear how many CpG sites are in the promoter and near vicinity, or why these 3 selected.

o    Information about the study sample is lacking – this is important since smoking and weight can influence methylation levels. In addition, no correction for age, weight or other possible confounders was performed. This should be justified or acknowledged as a limitation in the discussion. Was the DNA pooled? It is not mentioned here.

·     In the discussion the authors should be careful not to overstate the significance of the results. Without experimental evidence to demonstrate differences in gene expression this is early, exploratory analysis. Strengths and limitations should be discussed also.

·     Please include a data availability statement.

Comments on the Quality of English Language

Overall the standard of writing is high, but comprehension and clarity would be improved by avoiding long sentences.

Author Response

Dear reviewer! We are very grateful for the high assessment and review of the article, as well as for valuable comments and observations. We have tried to answer all the questions and make all the necessary corrections.

The subject of epigenetic regulation of bone metabolism is one of great interest and the investigation of differential methylation of RUNX2 and its potential role in the development of osteoporosis is worthy of study.

In the introduction the authors give a very good overview of the literature, although the structure could be improved, and missing is a clear rational for performing the study and for the selection of the three CpG sites studied. Improving the written descriptions and visual representations of the RUNX promoter region and location of the CpG sites in relation to other CPG sites, regulatory elements or SNPs would greatly improve the clarity. The discussion also provides a good overview of the literature but could be improved by contextualising and interpreting the results.

Specific comments include:

  1. The title is, I think, overstated and revision is suggested.

Answer: dear reviewer, with your permission, we suggest leaving the name in its original form, since it is based on an indication of the probabilistic (that is, potential, but not yet replicated) nature of the marker that affects the risk of developing the disease, therefore the word "potential" was used in the title.

  • Results:
  1. Some methods are included here and would be more appropriate, moved to the relevant section.

Answer: Dear reviewer, we fully agree with the comment and have made the necessary changes.

  1. Legends to figures and tables could be more detiled to aid understanding of the contents e.g. In table 2 what is the sample estimate?

Answer: Dear reviewer, we fully agree with the comment and have added additional information in the captions to the figures and in the note to the table.

  1. Figure 1 resolution is poor. Figure 4 is not very informative. Inclusion of a diagram illustrating the CpG sites in relation to key features would aid interpretation.

Answer: dear reviewer, we fully agree with the remark and completely redesigned the Figures for a more schematic and visual representation of the gene structure and localization of the studied CpG sites.

  • Methods:
  1. Description of the RUNX CpG sites, location etc should be included here to improve overall understanding. For example, it is unclear how many CpG sites are in the promoter and near vicinity, or why these 3 selected.

Answer: dear reviewer, we agree with the remark, therefore, we have added additional information about the selection criteria for selected CpG sites in the section "DNA methylation analysis", and also visually displayed in Figure 1 the localization of the desired CpG sites relative to exons and promoters of the RUNX2 gene.

  1. Information about the study sample is lacking – this is important since smoking and weight can influence methylation levels. In addition, no correction for age, weight or other possible confounders was performed. This should be justified or acknowledged as a limitation in the discussion. Was the DNA pooled? It is not mentioned here.

Answer: dear reviewer, we fully agree with the remark, therefore we have added additional information about weight, body mass index and exclusion of smokers from the sample. The sample is completely homogeneous and includes patients with severe osteoporosis and control sample people from the same weight range, body mass index, as well as age category, which proves the correctness of statistical analysis and study design. We also clarified the information in the "Results" section, where we added information about DNA methylation analysis in a combined sample of men and women.

  1. In the discussion the authors should be careful not to overstate the significance of the results. Without experimental evidence to demonstrate differences in gene expression this is early, exploratory analysis. Strengths and limitations should be discussed also.

Dear reviewer, we agree with your comments. We have added a section on the limitations and strengths of the study

  1. Please include a data availability statement.

Answer: dear reviewer, all the research data, including information about patients, data on the examination of the local bioethical committee, as well as data from the results of static calculations and summary data on the indicators of DNA methylation status are fully presented in the article.

Reviewer 2 Report

Comments and Suggestions for Authors

In this study, Yalaev et al. investigated the methylation levels of the Runt-related transcription factor 2 (RUNX2) in Russian osteoporosis patients and identified the presence of a CpG site which was significantly hypomethylated. Additionally, the authors analyzed the gene structure and pinpointed the location of the investigated DNA fragment, which is in the promoter region of the RUNX2 gene. The authors discussed the functional significance of the hypomethylation in osteoporosis. This is an interesting study. Their findings reveal a potential novel regulatory mechanism underlying the progression of osteoporosis. I have several comments and suggestions for the authors.

It is great to discover that the CpG site is hypomethylated in osteoporosis patients. It would be better if the authors validated the functional significance of the hypomethylation by introducing artificially hypomethylated and/or hypermethylated CpG sites in the RUNX2 promoter region and examining the bone phenotype using some bioresources such as mice.

This manuscript has several incorrectly abbreviated words. For example, Transcription factor 2 (RUNX2) in line 53. This is the first appearance of RUNX2 in the main text (not in abstract) and should be spelled out in full. I strongly suggest that the authors include a list of abbreviations used in this manuscript.

The following sentence does not make sense: NF-kB ligand receptor activator, which affects acetylation (RANKL) and (SOST) in line 254-255. What is “NF-kB ligand receptor activator”? I supposed this indicates “The receptor activator of NF-kappaB ligand (RANKL)”. Additionally, please name and cite references for the compounds that act on the epigenetic landscape of bone tissue remodeling balance.

Author Response

Dear reviewer! We are very grateful for the positive feedback, high appreciation and review of the article, as well as for valuable comments and observations. We have tried to answer all the questions and make all the necessary corrections.

In this study, Yalaev et al. investigated the methylation levels of the Runt-related transcription factor 2 (RUNX2) in Russian osteoporosis patients and identified the presence of a CpG site which was significantly hypomethylated. Additionally, the authors analyzed the gene structure and pinpointed the location of the investigated DNA fragment, which is in the promoter region of the RUNX2 gene. The authors discussed the functional significance of the hypomethylation in osteoporosis. This is an interesting study. Their findings reveal a potential novel regulatory mechanism underlying the progression of osteoporosis. I have several comments and suggestions for the authors.

  1. It is great to discover that the CpG site is hypomethylated in osteoporosis patients. It would be better if the authors validated the functional significance of the hypomethylation by introducing artificially hypomethylated and/or hypermethylated CpG sites in the RUNX2 promoter region and examining the bone phenotype using some bioresources such as mice.

Answer: dear reviewer, indeed, functional studies are an important component of comprehensive studies aimed at studying the influence of certain epigenetic markers on the risks of primary osteoporosis. Our study is a pilot and has scientific novelty, because for the first time we were able to identify statistically significant associations of differential DNA methylation of the RUNX2 gene promoter with the risk of osteoporosis, and also conducted a theoretical assessment of the functional significance of the identified changes in the pathogenesis of osteoporosis. Together, this allowed us to "get on the trail" and prepare the ground for the evidence base for the implementation of projects for further functional studies in vitro and in silicon, which, taking into account previously identified correlations, will allow for a more comprehensive and detailed analysis of the significance of hypomethylation of the RUNX2 gene in the pathogenesis of osteoporosis.

  1. This manuscript has several incorrectly abbreviated words. For example, Transcription factor 2 (RUNX2) in line 53. This is the first appearance of RUNX2 in the main text (not in abstract) and should be spelled out in full. I strongly suggest that the authors include a list of abbreviations used in this manuscript.

Answer: dear reviewer, we fully agree with the comment and made the necessary adjustments, and also added a sub-chapter with a list of abbreviations.

  1. The following sentence does not make sense: NF-kB ligand receptor activator, which affects acetylation (RANKL) and (SOST) in line 254-255. What is “NF-kB ligand receptor activator”? I supposed this indicates “The receptor activator of NF-kappaB ligand (RANKL)”. Additionally, please name and cite references for the compounds that act on the epigenetic landscape of bone tissue remodeling balance.

Answer: Dear reviewer, we fully agree with the comment and have made the necessary adjustments, as well as added new information about examples of compounds that affect the epigenetic regulation of bone metabolism.

Round 2

Reviewer 2 Report

Comments and Suggestions for Authors

I understand that this is the beginning of research into the implication of RUNX2 methylation in osteoporosis. Future research, including functional studies, will reveal the further significance of hypomethylation of RUNX2 in osteoporosis. I believe this manuscript has been sufficiently improved to be published in IJMS. I am looking forward to the authors’ subsequent studies.